# Multivariate Genomic Hybrid Prediction with Kernels and Parental Information

**DOI:** 10.3390/ijms241813799

**Published:** 2023-09-07

**Authors:** Osval A. Montesinos-López, José Crossa, Carolina Saint Pierre, Guillermo Gerard, Marco Alberto Valenzo-Jiménez, Paolo Vitale, Patricia Edwigis Valladares-Cellis, Raymundo Buenrostro-Mariscal, Abelardo Montesinos-López, Leonardo Crespo-Herrera

**Affiliations:** 1Facultad de Telemática, Universidad de Colima, Colima 28040, Colima, Mexico; oamontes2@hotmail.com (O.A.M.-L.); raymundo@ucol.mx (R.B.-M.); 2International Maize and Wheat Improvement Center (CIMMYT), Km 45, Carretera México-Veracruz, Texcoco 52640, México, Mexico; j.crossa@cgiar.org (J.C.); c.saintpierre@cgiar.org (C.S.P.); guillermosgerard@gmail.com (G.G.); p.vitale@cgiar.org (P.V.); 3Colegio de Postgraduados, Montecillos 56230, México, Mexico; 4Universidad Michoacana de San Nicolas de Hidalgo (UMSNH), Avenida Francisco J. Mujica S/N Ciudad Universitaria, Morelia 58030, Michoacán, Mexico; 5Bachillerato 22, Universidad de Colima, Cuauhtémoc 28510, Colima, Mexico; 6Centro Universitario de Ciencias Exactas e Ingenierías (CUCEI), Universidad de Guadalajara, Guadalajara 44430, Jalisco, Mexico

**Keywords:** hybrid prediction, parental information, integration, multi-trait

## Abstract

Genomic selection (GS) plays a pivotal role in hybrid prediction. It can enhance the selection of parental lines, accurately predict hybrid performance, and harness hybrid vigor. Likewise, it can optimize breeding strategies by reducing field trial requirements, expediting hybrid development, facilitating targeted trait improvement, and enhancing adaptability to diverse environments. Leveraging genomic information empowers breeders to make informed decisions and significantly improve the efficiency and success rate of hybrid breeding programs. In order to improve the genomic ability performance, we explored the incorporation of parental phenotypic information as covariates under a multi-trait framework. Approach 1, referred to as *Pmean*, directly utilized parental phenotypic information without any preprocessing. While approach 2, denoted as *BV*, replaced the direct use of phenotypic values of both parents with their respective breeding values. While an improvement in prediction performance was observed in both approaches, with a minimum 4.24% reduction in the normalized root mean square error (*NRMSE*), the direct incorporation of parental phenotypic information in the *Pmean* approach slightly outperformed the *BV* approach. We also compared these two approaches using linear and nonlinear kernels, but no relevant gain was observed. Finally, our results increase empirical evidence confirming that the integration of parental phenotypic information helps increase the prediction performance of hybrids.

## 1. Introduction

Meeting the increasing global demand for food is an imperative challenge, especially in the face of climate change and its subsequent impact on natural resources. Plant breeding plays a crucial role in the human food chain by contributing to high and stable production yields with minimal external inputs and environmental impact. In the last twenty years, plant breeding has been revolutionized by Genomic Selection (GS), a predictive methodology proposed by Meuwissen et al. (2001) [1], which has enabled the selection of superior candidates based solely on genotypic information.

GS is important because it allows breeders to make more accurate and efficient selection decisions in plant and animal breeding programs. It involves the use of genomic data, such as DNA markers, to predict the genetic value of an individual or a population for a given trait of interest. This prediction is based on the association between the genomic data and the phenotype (observable characteristics) of the individuals [2,3,4,5]. Compared to traditional breeding methods, GS has several advantages. Firstly, individuals with desired traits can be selected earlier, reducing the time and cost of breeding programs. Likewise, traits that are difficult to measure or are expressed late in the life cycle can also be selected. Additionally, GS can increase the genetic gain per unit of time and cost, leading to the development of more productive and resilient crops and livestock. Overall, GS has revolutionized the field of plant and animal breeding, making it more efficient and effective [6,7,8].

Nonetheless, the successful implementation of GS in plant breeding for predicting the performance of hybrid combinations based on the genetic makeup of their parents poses several challenges, which include: (1) Non-additive effects: Hybrid prediction assumes additive genetic effects of parental lines, but in reality, non-additive effects, such as dominance and epistasis, significantly influence hybrid performance. Accurately modeling non-additive effects requires larger sample sizes for estimation; (2) Genotype-by-environment (G × E) interaction: Hybrid performance varies across different environments due to G × E interaction. This variation makes it challenging to predict hybrid performance accurately across diverse environments; (3) Limited data: The availability of limited data poses difficulties in accurately predicting hybrid performance, especially for new or untested hybrid combinations. Insufficient data hinders accurate predictions of hybrid performance; (4) Heterogeneous parental populations: Genetic diversity among parental lines used in hybrid creation complicates the accurate prediction of hybrid performance. This challenge is particularly prominent when dealing with open-pollinated populations or composite crosses; and (5) Complex trait architecture: The accurate modeling of complex traits, such as yield or disease resistance, is difficult, posing challenges in predicting hybrid performance for these traits. In conclusion, hybrid prediction in plant breeding is a complex and challenging task that requires careful consideration of the underlying genetic and environmental factors influencing hybrid performance [9].

In this regard, a valuable alternative to genomic prediction in plant breeding is multi-trait hybrid prediction, which permits the simultaneous prediction of hybrid performance for multiple traits, reducing both time and cost in breeding programs. Multi-trait hybrid prediction is advantageous for several reasons, including (1) Improved accuracy: By considering trait correlations, multi-trait hybrid prediction enhances the accuracy of performance prediction. Leveraging similarities in the genetic basis of traits, information from one trait improves predictions for another; (2) Reduced bias: Traditional genomic prediction methods may exhibit bias due to an uneven distribution of phenotypic data or low heritability of traits. Multi-trait hybrid prediction mitigates such bias by integrating genetic information from multiple traits to estimate hybrid genetic value; (3) Enhanced hybrid selection: Multi-trait hybrid prediction facilitates the selection of hybrids with desired combinations of traits, leading to more productive and resilient crops. This is especially valuable for complex traits that are difficult to measure or expressed late in the life cycle; and (4) Improved management of G × E interaction: Multi-trait hybrid prediction assists breeders in effectively managing genotype-by-environment (G × E) interaction. It considers trait correlations and their interaction with environmental factors, aiding the identification of hybrids with consistent performance across diverse environments. Overall, multi-trait hybrid prediction shows promise in improving the accuracy and efficiency of plant breeding programs, resulting in the development of highly productive and resilient crops [10,11].

Montesinos-López et al. (2022) [12,13] showed that multi-trait prediction using the kernel method has the potential to enhance prediction accuracy as kernels are able to capture nonlinear patterns when they are in the data. Kernel methods are important for genomic prediction because they efficiently model complex, nonlinear relationships between genetic variants and phenotypes. In genomic prediction, the goal is to predict phenotypic traits based on genetic information, and kernel methods provide a powerful framework for achieving this. Kernel methods rely on the concept of a kernel function, which can be thought of as a measure of similarity between two objects. In the context of genomic prediction, the objects are typically genetic variants, and the kernel function measures the similarity between two variants based on their genetic similarity [13]. By using kernel methods, genomic prediction models can capture complex interactions between genetic variants that may not be easily captured by linear models. This is important because many phenotypic traits of interest are influenced by numerous genetic variants, each of which may have a small effect. Kernel methods allow for these subtle interactions to be captured, leading to more accurate predictions. Overall, kernel methods are a powerful tool for genomic prediction, as they efficiently model complex, nonlinear relationships between genetic variants and phenotypes [1,14,15].

Since the prediction of hybrids in plant breeding is very challenging, many approaches have been studied to increase prediction accuracy. For example, some authors, such as Xu et al. (2021) [16] and Wang et al. (2012) [17], proposed the incorporation as an input of parental information to increase the prediction accuracy of hybrids because it provides additional information about the genetic background of the offspring. The genetic makeup of the offspring is a combination of the genetic makeup of the two parents, and this information is used to improve predictions of the offspring’s performance. Other studies proposed to incorporate phenotypic data or environmental data, among others.

For this reason, to evaluate if incorporating phenotypic parental information as an additional input improves prediction accuracy in the context of multi-trait with kernel methods, we utilized a wheat dataset provided by the International Maize and Wheat Improvement Center (CIMMYT). We hypothesize that incorporating the phenotypic information of the parents as covariates in the multivariate genomic prediction model with kernels will improve the prediction accuracy of the GS methodology. In this sense, we aim to broaden our understanding of the genetics of the studied traits and maximize the efficiency of hybrid prediction by exploring the multivariate kernel approach.

## 2. Results

The results presented for each trait and for each year are compared with the three strategies of incorporating the parental phenotypic information (*NO_Cov*, *BV*, and *Pmean*) in each of the six kernels, AC_1, AC_2, AC_3 AC_4, GK, and Linear. These comparisons were carried out for each year and across years.

### 2.1. Trait DTF

The following Figure 1 shows the prediction performance of each kernel in each year and across years (Global). For each kernel in each year, we compared the three strategies of incorporating the parental phenotypic information (*NO_Cov*, *BV*, and *Pmean*).

In Figure 1, we can appreciate that in each year and across years, the best prediction performance was observed incorporating the parental phenotypic information (*BV* and *Pmean*). However, small differences are observed between the different kernels, and we can appreciate that in year 1 under the strategy *BV*, the best prediction performance was observed under a linear kernel that outperformed the remaining kernels by only 1.534%. While under the *Pmean* strategy, also in year 1, the linear kernel was the best regarding the remaining kernels but only by 0.706%. The same pattern was observed in years 2 and 3 under the *BV* strategy, where the best linear kernel outperformed the remaining kernels by 2.125% and 2.33%, respectively, while under the *Pmean* strategy, the linear kernel outperformed the other in years 2 and 3 by 0.062% and 0.106%, respectively (Figure 1).

Finally, across years (Global), we can appreciate that the linear kernel was also the best under both strategies of incorporating the phenotypic parental information (Figure 1). However, the gain in terms of *NRMSE* of the linear kernel regarding the other kernels was 1.982% under the *BV* strategy and 0.304% under the *Pmean* strategy. Under the linear kernel, we can appreciate that the best strategy for incorporating the phenotypic information (both for each year and across years) was the *Pmean* with an *NRMSE* = 0.626, followed by the *BV* strategy with an *NRMSE* = 0.632 and without incorporating parental phenotypic information. The *NO_Cov* strategy was the worst with an *NRMSE* = 0.666, meaning that the *Pmean* and *BV* strategies were better than the *NO_Cov* by 6.4% and 5.4%, respectively. See details in Table A1.

### 2.2. Trait DTH

Figure 2 demonstrates that including parental phenotypic information (*BV* and *Pmean*) consistently resulted in the most accurate predictions across different years. Although there were slight differences among various kernels, the linear kernel stood out in year 1, outperforming others by 1.577% using the *BV* strategy and 0.468% using the *Pmean* strategy. This trend continued in years 2 and 3 under the *BV* strategy, with the linear kernel leading by 1.875% and 2.448%, respectively. In year two, under the *Pmean* strategy, all six kernels performed equally, while in year three, the linear kernel was slightly better, with a 0.142% advantage.

When considering all years together (Global), the linear kernel stood out as the top performer in both the *BV* and *Pmean* strategies, improving *NRMSE* by 1.955% and 0.202%, respectively, compared to other kernels. Interestingly, the best approach for integrating phenotypic information, represented by the linear kernel, was the *Pmean* strategy, resulting in an *NRMSE* of 0.610. The *BV* strategy followed closely with an *NRMSE* of 0.617. In contrast, the strategy of not including parental phenotypic information (*NO_Cov* strategy) performed the worst, with an *NRMSE* of 0.651. This means that the *Pmean* and *BV* strategies outperformed the *NO_Cov* strategy by 6.7% and 5.5%, respectively. For more detailed information, please refer to Table A2.

### 2.3. Trait YIELD

Figure 3 illustrates that including parental phenotypic data (*BV* and *Pmean*) consistently led to the best predictive performance across different years. Nevertheless, there were subtle differences among the various kernels. Specifically, in year one, the linear kernel performed slightly worse than the other kernels by only 0.099% when using the *BV* strategy. Meanwhile, under the *Pmean* strategy, the linear kernel performed equally as well as the other kernels. This pattern persisted in years 2 and 3 under the *BV* strategy, where the linear kernel showed slightly lower performance than the other kernels by 0.075% and 0.194%, respectively. Similarly, under the *Pmean* strategy, the linear kernel exhibited slightly lower performance than the other five kernels, but the difference was minimal, with only 0.013% and 0.071% variations in years two and three, respectively.

When considering all years collectively (Global), the linear kernel showed a slight performance disadvantage compared to other kernels in both the *BV* and *Pmean* strategies, resulting in a decrease in *NRMSE* of 0.120% and 0.026% relative to the other kernels (Figure 3). Interestingly, the best strategy for incorporating phenotypic information, as indicated by the top-performing kernels (AC_1, AC_2, AC_3, AC_4, and GK), was the *Pmean* strategy, which achieved an *NRMSE* of 0.766. The *BV* strategy followed closely with an *NRMSE* of 0.778. In contrast, the strategy of not incorporating parental phenotypic information (*NO_Cov* strategy) showed the poorest performance, with an *NRMSE* of 0.796 (Figure 3). This demonstrates that the *Pmean* and *BV* strategies outperformed the *NO_Cov* strategy by 3.9% and 2.3%, respectively. For more detailed information, please refer to Table A3.

### 2.4. Across Traits

Since the linear kernel was the best in two out of three of the traits under study, in this section, we present the results across traits only for this linear kernel.

Across traits and years, we can appreciate in Figure 4 that not incorporating the parental phenotypic information as a covariate in the modeling process decreases prediction accuracy by 4.24% when the parental phenotypic information is incorporated as breeding values (*BV*) estimated from the parents, and by 5.59% when the parental phenotypic information is incorporated directly as the blues of the parents. In other words, these results point out that incorporating the phenotypic information helps to increase the prediction accuracy of hybrid predictions by at least 4.24% under a multi-trait framework.

## 3. Discussion

Hybrid prediction is of utmost importance in plant breeding as it harnesses the benefits of hybrid vigor, improves yield potential, enhances genetic diversity, facilitates trait selection, optimizes resource utilization, accelerates breeding progress, increases crop productivity, promotes sustainable agriculture, addresses consumer preferences, and provides economic advantages to stakeholders. However, as mentioned in the introduction, many factors affect the efficient development of highly productive hybrids.

For these reasons, the GS methodology is crucial for hybrid development as it enhances breeding efficiency, improves prediction accuracy, enables early-stage selection, aids in complex trait prediction, manages genetic diversity, facilitates information transferability, complements marker-assisted selection, enables selection for novel traits, supports data-driven breeding decisions and can be integrated with other breeding approaches [1,9]. By leveraging genomic information, breeders can optimize hybrid breeding programs and accelerate the development of high-performing and genetically superior hybrids.

However, for a successful implementation of the GS methodology, high accuracy is key for an efficient selection, increasing genetic gain, saving costs and time, utilizing genetic resources effectively, selecting for complex traits, facilitating precision breeding, building confidence in breeding decisions, adapting to changing years, managing genetic diversity, and promoting industry acceptance and adoption. Achieving high accuracy in genomic selection enhances the effectiveness and efficiency of plant breeding programs, ultimately leading to the development of improved and high-performing hybrids.

For this reason, the prediction performance of the GS methodology was explored under a multivariate framework with two strategies for incorporating the parental phenotypic information in the modeling process as covariates. The first approach, denoted as Pmean, directly used the parental phenotypic information without any preprocessing, while the second, denoted as BV, used only the breeding values of the parents ($g_{M,t}$ and $g_{F,t}$) instead of the phenotypic values of both parents ($P_{M,t}$ and $P_{F,t}$). Under both approaches, an increase in prediction performance NRMSE of at least 4.24% was observed; however, the direct approach of incorporating the parental phenotypic information (Pmean) was slightly better than the BV approach. However, we do not have elements to say that the Pmean approach is statistically better than the BV approach; for this reason, the small difference between the two approaches can be attributed in part to Monte-Carlo error since we implemented both approaches under a Bayesian framework.

Likewise, under both strategies (Pmean and BV) of incorporating the parental phenotypic information, we explored the use of nonlinear inputs using different kernels (AC_1, AC_2, AC_3, AC_4, and GK), which were compared to the conventional linear kernel. We did not find relevant differences in prediction performance between the kernels using the NRMSE as a metric. As such, findings suggest that, in general, for this data set, the linear kernel is sufficient, for when nonlinear kernels were evaluated, no significant gain in prediction performance was observed. However, even though the nonlinear kernels were not better in terms of NRMSE than the conventional linear kernel, it is of paramount importance to remember that in many data sets, the use of these nonlinear kernels still helps to increase prediction performance since they can efficiently capture nonlinear patterns in the input data when they are present.

In general, genomic prediction models under a multivariate context with nonlinear kernels have the potential to capture complex relationships, improve prediction accuracy, consider trait correlations, account for genetic pleiotropy and interactions, uncover hidden patterns, offer flexibility and adaptability, allow for cross-species applications, support better breeding decisions, and contribute to advancements in data science. By incorporating these models, breeders can enhance the accuracy and efficiency of genomic prediction, leading to improved plant breeding outcomes and the development of superior hybrids.

However, the efficacy of the multivariate model in comparison to the single-trait analysis varies depending on the specific problem at hand. Conventional multivariate models presuppose a uniform covariance of effects throughout the genome. In genomic regions where the alignment of effects correlates closely with the average effect correlation across the genome, leveraging information sharing among traits can enhance statistical power. Conversely, in regions where the correlation of effects significantly deviates from the genome’s average correlation pattern, the multivariate model might diminish statistical power. This reduction can be attributed to the tendency of multivariate models to constrict the magnitude of effects towards a shared covariance pattern [18].

Finally, our findings increase empirical evidence that integrating parental phenotypic information improves prediction performance by integrating this information as covariates. It is also important to point out that we did not find any improvement when adding this information under a multi-trait framework versus a uni-trait framework. For this reason, by incorporating parental phenotypic information, prediction models can make more accurate predictions and support more effective breeding decisions in plant breeding programs. However, we did not find an improvement in integrating this information as BV as opposed to integrating it as Pmean.

## 4. Materials and Methods

### 4.1. Phenotypic Data

Field experiments were conducted at CIMMYT’s Campo Experimental Norman E. Borlaug (CENEB) near Ciudad Obregon, Sonora, Mexico, over a period of three years. A total of 1888 hybrids resulting from crosses between 667 females and 18 males were evaluated. Specifically, the number of hybrids assessed during the winter growing seasons of 2014–2015 (Year 1), 2015–2016 (Year 2), and 2016–2017 (Year 3) were 703, 655, and 1197, respectively. Among these, 225 hybrids were common between consecutive years (Years 1 and 2), while 383 hybrids were common between Years 2 and 3. The selection of elite female and male parents was based on their performance for desired traits, ability to produce hybrids, and ancestral diversity, which was determined using a coefficient of parentage.

In order to produce the hybrids, a chemical hybridizing agent provided by Syngenta Inc. (Wilmington, DE, USA) was utilized in alternate male and female strip plots measuring 6.4 m. The parents and hybrids were evaluated in α-lattice trials, with two replications conducted over a span of two years. Each 4.8 m yield trial plot consisted of 1000 seeds to ensure uniform plant density. Standard agronomic practices, including four supplementary irrigations, were employed in a high-yield-potential environment. All male parents involved in the hybrids and the set of two advanced checks tested each year were planted in all trials. The hybrids and female parents were planted side by side in all the trials. For each entry, data on days to flowering (DTF), days to heading (DTH), days to maturity (DTM), grain yield (GY), and plant height (PHT) per plot were recorded. Phenotypic data were analyzed using a mixed linear model implemented in META-R software (V5.0). Best linear unbiased estimates (BLUEs) were estimated by fitting the model with trial (as a random effect), genotype (as a fixed effect), replication nested within trials (as a random effect), and sub-blocks nested within trials and replications (as random effects). The obtained BLUEs for each hybrid and parent were utilized for subsequent analyses. This paper focuses on the analysis of three traits: grain yield (GY), days to flowering (DTF), and days to heading (DTH).

### 4.2. Genotypic Data

In the first year, 18 male and 667 female parents underwent genotyping using the Illumina iSelect 90K Infinitum SNP genotyping array. In the second and third years, genotyping was performed using the Illumina Infinium 15K wheat SNP array (TraitGenetics GmbH, Gatersleben, Germany). After combining the data from all three years, a total of 13,005 single-nucleotide polymorphisms (SNPs) remained. SNPs with less than 15% missing values were retained, and any remaining missing markers were imputed using the naive method based on observed allele frequencies. Following imputation, markers with a minor allele frequency below 0.05 were excluded from the analysis. A total of 10,250 markers were ultimately utilized for further analysis. Although a larger set of hybrids and parents were assessed in the field experiments, only hybrids derived from parents that had undergone SNP genotyping were considered for genomic predictions. The number of hybrids included varied in each year of evaluation.

### 4.3. Multivariate Statistical Model

This model is given by
(1)$$Y=Z_{E}\beta_{E}+Z_{M}g_{M}+Z_{F}g_{F}+Z_{H}h+u_{M}+u_{F}+u_{H}+X_{\mathrm{AC}}\beta_{\mathrm{AC}}+\epsilon$$
where Y is the matrix of response variables of order $n\times n_{T}$ (with $n_{T}=3$ since the traits under study were GY, DTF, and DTH); n denotes the total number of observations; $Z_{E}$ is the design matrix for environments (year); $\beta_{E}$ is the matrix of year effects of order $I\times n_{T}$, and I denotes the number of years, and it is assumed as random effects since model (1) was implemented under a Bayesian framework; $\beta_{E}$ $∼{\mathrm{MN}}_{I\times n_{T}}\left(0,\sigma_{E}^{2}I_{I},I_{n_{T}}\right)$, that is, with a matrix-variate normal distribution with parameters M=0, $U=\sigma_{E}^{2}I_{I}$ and $V=I_{n_{T}}$, and $g_{M}$ is the matrix of random effects due to the general combining ability (GCA) of markers from paternal lines (males, M); $g_{F}$ is the matrix of random effects due to the GCA of markers for maternal lines (females, F), and h is the matrix of SCA random effects for the crosses (hybrids, H). The incidence matrices $Z_{M}$, $Z_{F}$, and $Z_{H}$ relate Y to $g_{M}$, $g_{M}$, and h with $g_{M}∼{\mathrm{MN}}_{M\times n_{T}}\left(0,G_{M},\Sigma_{M}\right)$, M denotes the male parents; $g_{F}∼{\mathrm{MN}}_{F\times n_{T}}\left(0,G_{F},\Sigma_{F}\right)$, F denotes the female parents, and $h∼{\mathrm{MN}}_{H\times n_{T}}\left(0,H=Z_{M}G_{M}Z_{M}^{T}⨀Z_{F}G_{F}Z_{M}^{T},\Sigma_{H}\right)$, H denotes the hybrids resulting from combining the M males and F females, ⨀ denotes the Hadamard product, where, $\Sigma_{M}$, $\Sigma_{F}$, and $\Sigma_{H}$ are variance-covariance components associated with GCA and SCA, and $G_{M}$, $G_{F}$, and H are relationship matrices for parental and maternal lines and hybrids, respectively. While $u_{M}∼{\mathrm{MN}}_{\mathrm{ME}\times n_{T}}\left(0,V_{M},\Sigma_{M_{E}}\right),$ denotes the random effects of males-year combinations, $u_{F}∼{\mathrm{MN}}_{\mathrm{FE}\times n_{T}}\left(0,V_{F},\Sigma_{M_{F}}\right)$, denotes random effects of females-year combinations, $u_{H}∼{\mathrm{MN}}_{\mathrm{HE}\times n_{T}}\left(0,V_{H},\Sigma_{M_{H}}\right)$; denotes the random effects of hybrids-year combinations, $\Sigma_{M_{E}}$, $\Sigma_{M_{F}}$, and $\Sigma_{M_{H}}$ are variance-covariances matrices of components associated with male × year, female × year, hybrid × year interactions, respectively; and $V_{M}$, $V_{F}$, and $V_{H}$ are the associated variance–covariance matrices. These variance-covariance matrices are given by $V_{M}=Z_{M}G_{M}Z_{M}^{T}⨀Z_{E}Z_{E}^{T}$, $V_{F}=Z_{F}G_{F}Z_{F}^{T}⨀Z_{E}Z_{E}^{T}$ and $V_{H}=Z_{H}{\mathrm{HZ}}_{H}^{T}⨀Z_{E}Z_{E}^{T}$. Finally, ϵ is the residual matrix of dimension $n\times n_{T}$ distributed as $\epsilon ∼{\mathrm{MN}}_{n\times n_{T}}\left(0,I_{n},R\right)$, where R is the residual variance-covariance matrix of order $n_{T}\times n_{T}$.

The relationship matrices $G_{M}$ and $G_{F}$ were computed using markers (Van Raden, 2008). Let $X_{m}$, m∈ {Male, Female} be the matrix of markers and let $W_{m}$, be the matrix of centered and standardized markers. Then $G_{m}=\frac{W_{m}W_{m}^{T}}{p}$ [5,19] where p is the number of markers. $X_{\mathrm{AC}}$ is the matrix that contains the parental covariates of the trait to be predicted and of correlated traits. The parental information was used under two approaches. The first one (Pmean) directly used the parental phenotypic information without any preprocessing. Under this approach, from each trait, we computed two covariates using the parental phenotypic information. One covariate that captured the additive part is computed as
(2)$$X_{\mathrm{AC},t,a}=\frac{\left(P_{M,t}+P_{F,t}\right)}{2}$$
where $P_{M,t}$ denotes the male parental phenotype for trait t, $P_{F,t}$ denotes the female parental phenotype for trait t, with t=GY,DTF, and DTH; while a denotes the additive effects. The other covariates captured the dominance part, and it is computed as
(3)$$X_{\mathrm{AC},t,d}=\frac{\left|P_{M,t}-P_{F,t}\right|}{2}$$
where d denotes dominance. The matrix $X_{\mathrm{AC}}$ contains six columns since two covariates (one for a and the other for d) were computed for each of the three traits under study. The second approach is denoted as BV. To incorporate the parental phenotypic information in place of using directly the phenotypic values of both parents ($P_{M,t}$ and $P_{F,t}$) we used the breeding values of the parents ($g_{M,t}$ and $g_{F,t}$) and these were computed with the following predictor:(4)P=1μ+g+e
where P is the vector of the response variable for each trait of order $n_{p}\times 1$, 1 is a vector of ones of order $n_{p}\times 1$, μ denotes a general mean, g is the random effects of both parental lines distributed as $g∼N_{n_{p}\times 1}\left(0,G\right)$, with G representing the genomic relationship matrix, and e are the residual errors distributed as $g∼N_{n_{p}\times 1}\left(0,I_{n_{p}}\sigma_{e}^{2}\right)$. After fitting model (4), we estimated the breeding values, g, that contain the breeding values of males ($g_{M,t})$ and females ($g_{F,t}$). Then, with these breeding values, we computed the covariates $X_{\mathrm{AC},t,a}$ and $X_{\mathrm{AC},t,d}$ with Equations (2) and (3), but instead of using $P_{M,t}$ and $P_{F,t}$, $g_{M,t}$ and $g_{F,t}$ were used. Finally, to compare both approaches for using the phenotypic information, Pmean and BV were evaluated with and without the phenotypic information as covariates. For this reason, three strategies resulted in incorporating or ignoring the parental phenotypic information. These strategies are the Pmean, the BV, and the NO_Cov method that ignored the parental phenotypic information. The implementation of these models was carried out in the R statistical software (V.6) using the BGLR library [20].

### 4.4. Evaluation of Prediction Performance

In each of the three methods evaluated (Pmean, BV, and NO_Cov), we employed a type of cross-validation that emulates real breeding strategies, referred to as untested lines in tested years, using a seven-fold cross-validation [13]. In this approach, the training set was allocated to 7-1 folds, while the remaining fold was assigned to the testing set. This process was repeated until each of the seven folds had been utilized at least once in the testing set. The average performance across the seven folds was then reported as the prediction performance, using the normalized root mean square error (NRMSE) as the evaluation metric. In order to compare the prediction accuracies between models of the same type (Pmean and BV), the relative efficiencies in terms of NRMSE were computed as follows:$${\mathrm{RE}}_{\mathrm{NRMSE}}=\frac{{\mathrm{NRMSE}}_{\mathrm{Mx}}}{{\mathrm{NRMSE}}_{\mathrm{Mx}\_z}}$$
where ${\mathrm{NRMSE}}_{\mathrm{Mx}}$ and ${\mathrm{NRMSE}}_{\mathrm{Mx}\_z}$ denote the NRMSE of model x=Pmean and BV and z=NO_Cov respectively. if ${\mathrm{RE}}_{\mathrm{NRMSE}}>1,$ the best prediction performance in terms of NRMSE was obtained using method Mx_z, but when ${\mathrm{RE}}_{\mathrm{NRMSE}}<1,$ the best method was Mx. When ${\mathrm{RE}}_{\mathrm{NRMSE}}=1,$ both methods were equally efficient.

### 4.5. Kernel Methods

Kernel functions (kernel methods or kernel tricks) are mathematical functions used in various machine learning algorithms. These functions enable the algorithms to operate in a high-dimensional feature space without explicitly computing the coordinates of the data points in that space [15]. Kernel functions could uncover complex, nonlinear relationships between data points by projecting them into a higher-dimensional space, where the relationships become more apparent and easier to separate. This process is called the “kernel trick”, which avoids the explicit computation of the higher-dimensional feature space, saving computational resources and memory.

The kernel functions efficiently represent data and perform the kernel trick where the kernel functions take pairs of data points in the original space and calculate the inner product (similarity) between them in a higher-dimensional space. A popular kernel function is the linear one that, in genomic prediction, is basically represented by the Genomic Best Linear Unbiased Predictor (GBLUP). The nonlinear Gaussian kernel (GK) function, also known as the radial basis function kernel, depends on the Euclidean distance between the original attribute value vectors rather than on their dot product, $K\left(x_{i},x_{j}\right)=e^{-{\gamma \|x}_{i}-x_{j}\|{}^{2}}$ The Gaussian kernel method is very popular, but it is sensitive to the choice of the γ parameter and may be prone to overfitting.

The Arc-cosine (AC) kernel function uses the idea of forming artificial neural networks with more than one hidden layer (l). Cho and Saul [21] proposed a recursive relationship of repeating l times the interior product. It is important to point out that this kernel method is like a deep neural network since more than one hidden layer can be used. In this study, we have represented the AC with 1,2,3,4 hidden layers as AC_1, AC_2, AC_3, and AC_4, respectively.

## 5. Conclusions

In this paper, the integration of parental phenotypic information under a multi-trait framework with different kernels was explored. We evaluated two approaches for the integration of parental phenotypic information. We found an increase in prediction performance in the normalized mean square error of at least 4.24% by integrating the parental phenotypic information, but no relevant differences were observed between the two approaches for integrating the parental phenotypic information. Furthermore, we did not find a significant increase in prediction performance using nonlinear kernels regarding linear kernels. Finally, our findings increase empirical evidence that integrating parental phenotypic information as covariates helps to increase the prediction performance of hybrids prediction.

## Figures and Tables

**Figure 1 ijms-24-13799-f001:**
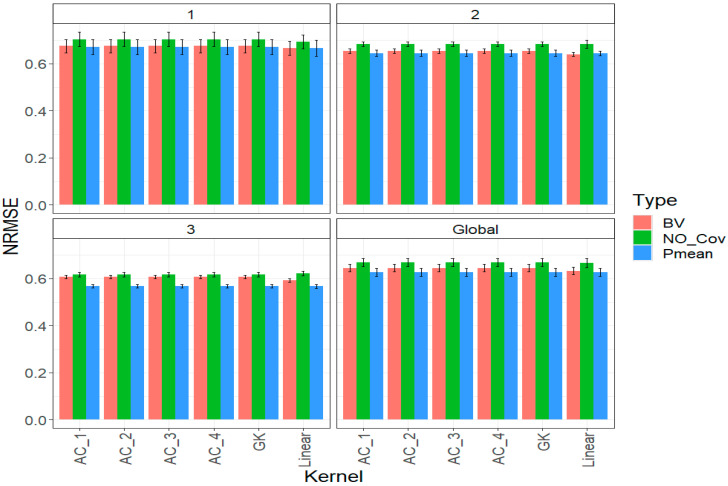
Prediction performance for trait DTF in each strategy of incorporating the parental phenotypic information (*NO_Cov*, *BV*, and *Pmean*) with six different kernel methods (AC_1, AC_2, AC_3, AC_4, Gaussian Kernel (GK) and linear kernel measured by *NRMSE* in each year (1, 2, 3) and across years (Global).

**Figure 2 ijms-24-13799-f002:**
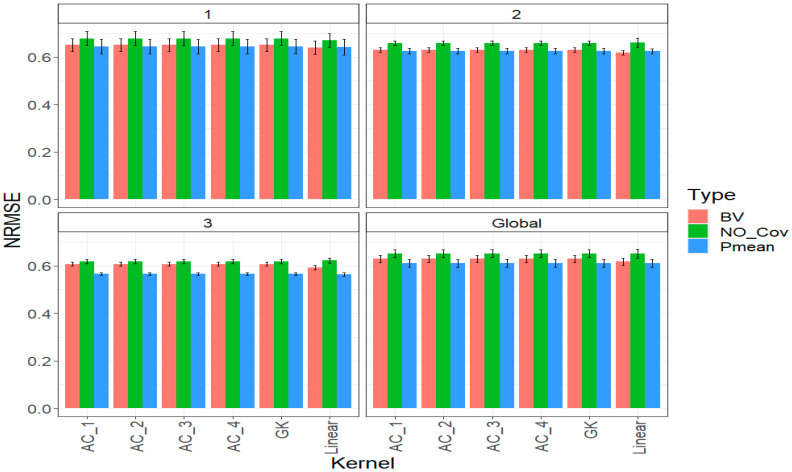
Prediction performance for trait DTH in each strategy of incorporating the parental phenotypic information (*NO_Cov*, *BV*, and *Pmean*) with six different kernel methods (AC_1, AC_2, AC_3, AC_4, Gaussian Kernel (GK) and linear kernel measured by *NRMSE* in each year (1, 2, 3) and across years (Global).

**Figure 3 ijms-24-13799-f003:**
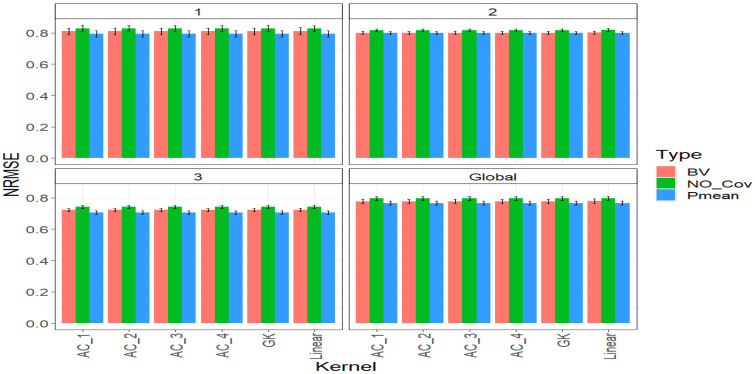
Prediction performance for trait GY in each strategy of incorporating the parental phenotypic information (*NO_Cov*, *BV*, and *Pmean*) with six different kernel methods (AC_1, AC_2, AC_3, AC_4, Gaussian Kernel (GK) and linear kernel measured by *NRMSE* in each year (1, 2, 3) and across years (Global).

**Figure 4 ijms-24-13799-f004:**
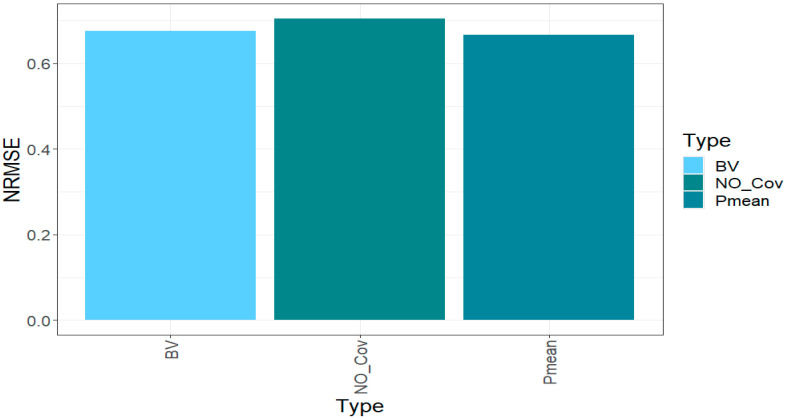
Prediction performance across traits and years for each strategy of incorporating the parental phenotypic information (*NO_Cov*, *BV*, and *Pmean*) with the linear kernel in terms of normalized root mean square error (*NRMSE*).

## Data Availability

Phenotypic and genomic data can be downloaded from the link: http://hdl.handle.net/11529/10548129.

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
