# Peer review of "Multivariate Genomic Hybrid Prediction with Kernels and Parental Information"

_ijms, 2023, doi:10.3390/ijms241813799_

Round 1

Reviewer 1 Report

This paper used three years wheat dataset to investigate the prediction efficiency for hybrid progeny by incorporating phenotypic parental information either as phenotypic mean or BV or none. This paper offers valuable insights into the realm of hybrid prediction and genomic selection. The exploration of parental phenotypic information's integration and its impact on prediction performance makes it a relevant read for researchers in the field. But I have a few concerns or comments for authors to address as below:

1)      In both introduction and discussion, some information is redundant or repetitively expressed, and it can be combined for a clearer flow. Make sure the introduction concludes with a clear lead into the methods or the main body of the paper, ensuring that the reader understands the research question and its significance.

2)      In Results session, there's a lot of repetition, especially when discussing how the linear kernel performed across different years and strategies. While it's important to detail differences, it can be presented in a more condensed manner.

3)      In Discussion session, repetitions can be reduced. For instance, the benefits of GS methodology are listed multiple times in different ways.

4)      It might also be worthwhile to explore why the Pmean approach outperformed BV slightly, even if the margin is small.

5)      Could authors explain why no improvement when comparing multi-trait versus uni-trait? What genetic correlations among those three traits?

6)      Line 229: “Genomic best linear unbiased predictions (BLUEs) …”, Did you fit GRM to get BLUEs? Was genotype fitted as random term?

7)      Line81: ‘many approaches have been studied have been ’ ?

8)      Line252: ‘andh’ ?

9)      Line270: ‘DTFandDTH,anda’ ?

10)   Line278: What does G mean ?

The draft tends to repeat certain concepts multiple times. For instance, the benefits and importance of hybrid prediction and the GS methodology are mentioned on several occasions. Reducing this repetition would make the text more concise and easier to follow.

Also please see above comments. 

Author Response

RESPONSE TO REVIEWER 1

This paper used three years wheat dataset to investigate the prediction efficiency for hybrid progeny by incorporating phenotypic parental information either as phenotypic mean or BV or none. This paper offers valuable insights into the realm of hybrid prediction and genomic selection. The exploration of parental phenotypic information's integration and its impact on prediction performance makes it a relevant read for researchers in the field. But I have a few concerns or comments for authors to address as below:

RESPONSE: Many thanks for your time invested in reading, revising and commenting the article. Authros truly appreciate and value your comments.

  • In both introduction and discussion, some information is redundant or repetitively expressed, and it can be combined for a clearer flow.Make sure the introduction concludes with a clear lead into the methods or the main body of the paper, ensuring that the reader understands the research question and its significance.

RESPONSE: Correction done in the new version of the paper. See lines 80-81 and 170.

  • In Results session, there's a lot of repetition, especially when discussing how the linear kernel performed across different years and strategies. While it's important to detail differences, it can be presented in a more condensed manner.

RESPONSE: yes thanks. We rephrased many paragraphs to improve the writing quality. See lines 123-133 and 140-153.

3)      In Discussion session, repetitions can be reduced. For instance, the benefits of GS methodology are listed multiple times in different ways.

RESPONSE: See correction on line 170.

  • It might also be worthwhile to explore why the Pmean approach outperformed BV slightly, even if the margin is small.

RESPONSE: Corrections and comments added  on. lines 186-188.

  • Could authors explain why no improvement when comparing multi-trait versus uni-trait? What genetic correlations among those three traits?

RESPONSE: Correction done in the new version of the paper. See lines 202-207.

6)      Line 229: “Genomic best linear unbiased predictions (BLUEs) …”, Did you fit GRM to get BLUEs? Was genotype fitted as random term?

RESPONSE: Yes indeed. See lines 229-231.

7)      Line81: ‘many approaches have been studied have been ’ ?

RESPONSE: Thanks for your point.. See lines 80-81.

8)      Line252: ‘andh’ ?

RESPONSE Done. See lines 252-253.

9)      Line270: ‘DTFandDTH,anda’ ?

 RESPONSE: Correction done in the new version of the paper. See line 270.

10)   Line278: What does G mean ?

RESPONSE; Clarified. See lines 278-279.

Comments on the Quality of English Language

The draft tends to repeat certain concepts multiple times. For instance, the benefits and importance of hybrid prediction and the GS methodology are mentioned on several occasions. Reducing this repetition would make the text more concise and easier to follow.

RESPONSE Yes we have reduced the repetition. See lines 80-81 and 170.

Reviewer 2 Report

I enjoyed reading this fine manuscript.  I particularly appreciated the use of cross-validation and thereby a distinctively predictive measure of performance.  I have no suggestions for improvement.

Author Response

RESPONSE TO REVIEWER 2

I enjoyed reading this fine manuscript.  I particularly appreciated the use of cross-validation and thereby a distinctively predictive measure of performance.  I have no suggestions for improvement.

RESPONSE: Thank you very much for your time reading this article.

Reviewer 3 Report

In line 244, there are no explanation for  5th to 7th elements in the equation.

In line 247, please explain why the effects for years follow normal distributions rather than fixed effects.

In 289, the authors mentioned the cross-validation was used in this study. Do you think if it is necessary to use replicates to make to results more reliable?

In line 292, do you think the correlation coefficient is a better way to measure prediction accuracy?

In the figures, I think it is better to delete the grey lines in the background.

Do you think if it is better to add more prediction methods in the comparison part?

Minor editing of English language required.

Author Response

RESPONSE TO REVIEWER 3

Open Review

In line 244, there are no explanation for  5th to 7th elements in the equation.

RESPONSE: Explanations are now given. See lines 250-255.

In line 247, please explain why the effects for years follow normal distributions rather than fixed effects.

RESPONSE: Correction done in the new version of the paper. See lines 246-248.

In 289, the authors mentioned the cross-validation was used in this study. Do you think if it is necessary to use replicates to make to results more reliable?

RESPONSE: In general, the more replicates for cross validation the more reliable the results, but of course this increase significantly the computational resources. For this reason, here we used 7-fold cross validation and each time one fold was used as testing set and the remaining six folds as training set until each fold was used as testing set one time. Finally, the average of the seven folds was reported as prediction performance. For this particular application we did not used more replications due to the lack of computational resources. However,  based on experience we not expect to obtain to a different conclusion in case of using more replications for the strategy of cross-validation.

In line 292, do you think the correlation coefficient is a better way to measure prediction accuracy?

RESPONSE: All metrics have pros and cons and the same applied for Pearson correlation. In genomic plant breeding  the Pearson´s correlation is the most widely metric for assessing genomic-enabled prediction accuracy..

In the figures, I think it is better to delete the grey lines in the background.

RESPONSE: Much appreciated. However, from the reader perspective we believe the grey lines help the readers to differentiate better the prediction performance between methods. We preferred to keep these lines.

Do you think if it is better to add more prediction methods in the comparison part?

RESPONSE: Thank you for question. We do not it is necessary to add another method. From our experience we have observed that in general the Gaussian kernel is one of the best methods to capture non-linear patterns and this method was evaluated. Also, we have observed that the GBLUP model which was used here is also very competitive and many times better than many state-of-the-art machine learning methods. In this article we used the GBLUP with the linear kernel and other non-linear kernel methods like Gaussian, Arcosine_1, …Arcosine_4).
